# Associations of body mass index, physical activity and sedentary time with blood pressure in primary school children from south-west England: A prospective study

Emma Solomon-Moore[1¤], Ruth Salway[1], Lydia Emm-Collison[1]*, Janice L. Thompson[2], Simon J. Sebire[1], Deborah A. Lawlor[3,4], Russell Jago[1]

1 Centre for Exercise, Nutrition & Health Sciences, School for Policy Studies, University of Bristol, Bristol, United Kingdom, 2 School of Sport, Exercise and Rehabilitation Sciences, University of Birmingham, Birmingham, United Kingdom, 3 Medical Research Council Integrative Epidemiology Unit, University of Bristol, Bristol, United Kingdom, 4 Population Health Science, Bristol Medical School, University of Bristol, Bristol, United Kingdom

¤ Current address: Department for Health, University of Bath, Bath, United Kingdom
* Lydia.Emm-Collison@bristol.ac.uk

## Abstract

Elevated blood pressure in children is a significant risk factor for the development of cardio-vascular disease in adulthood. We examined how children's body mass index (BMI), physical activity and sedentary time at ages 9 and 11 are associated with blood pressure at age 11. Data were from 1283 children from Bristol, UK, who participated in the study aged 11 years, 797 of whom also participated in the study aged 9 years. Child height, weight and blood pressure were measured, and children wore accelerometers for five days, from which moderate-to-vigorous-intensity physical activity and sedentary minutes per day were derived. Multiple imputation of missing data and adjusted linear and logistic regression models were used to examine associations. Child BMI at 11 years was cross-sectionally associated with higher systolic and diastolic blood pressure (mean difference [95% confidence interval]: 0.91 [0.32 to 1.50] mm Hg and 1.08 [0.54 to 1.62] mm Hg, respectively, per standard deviation (SD) of BMI). BMI at age 9 was also positively associated with diastolic blood pressure at age 11 (1.16 mmHg per two years [0.49 to 1.84], per SD of BMI). For girls, sedentary time at age 9 years was associated with increased odds of having high systolic blood pressure at age 11 (odds ratio: 1.08 [1.01 to 1.16], per 10 minutes per day). There was no evidence of associations between sedentary time and blood pressure among boys. Similarly, there was little evidence that physical activity was associated with blood pressure in either cross-sectional or prospective analyses. Effective strategies are needed to prevent excess bodyweight among children in order to reduce cardiovascular disease risk.

**Data Availability Statement:** All data files are available from the Zenodo database (DOI is 10.5281/zenodo.1049587)

**Funding:** This work was supported by grants from the British Heart Foundation (ref PG/11/51/28986 and SP 14/4/31123). DAL works in a unit that receives funding from the University of Bristol and UK Medical Research Council (MC_UU_00011/6); she is also a UK National Institute of Health and Research Senior Investigator (NF-0616-10102). The funders had no role in study design, data collection and analysis, decision to publish, or preparation of the manuscript.

**Competing interests:** The authors have declared that no competing interests exist.

## Introduction

Elevated blood pressure in children is a significant risk factor for the development of cardiovascular disease in adulthood. Increasing numbers of children and young people are being diagnosed with hypertension [1–3], with global prevalence rates of at least 3% among asymptomatic children and adolescents [2]. Higher childhood blood pressure tracks through to adulthood [4] and is positively associated with the development of cardiovascular disease later in life and risk of premature mortality [5–10]. Globally, children are also more likely to be categorised as overweight or obese than ever before [11, 12]. In 2017–2018, approximately one third of eleven-year-olds in England were categorised as overweight, while one fifth were categorised as obese [11]. Children whose overweight and obesity persist into adulthood are at an increased risk of hypertension, type 2 diabetes, dyslipidemia, and carotid-artery atherosclerosis compared to children who are normal weight [13]. Available evidence has generally demonstrated strong positive associations between body mass index (BMI) and blood pressure in children and adolescents [14–24]. However, the majority of studies were cross-sectional [15–19] or examined prospective associations with blood pressure measured at only a single time-point [21–24]. Prospective studies with blood pressure measures at baseline and follow-up, therefore enabling adjustment for baseline blood pressure, would be valuable to gain a better understanding of whether higher BMI is a risk factor for higher blood pressure in children.

Trend data from the UK suggests that BMI explained ~15% of the increases in systolic blood pressure between 1980–2008, with associations weakening over time [14]. Thus, there is value in examining other factors that may be associated with the rising levels of blood pressure in children in recent years. Physical inactivity is positively associated with higher blood pressure and cardiovascular disease in adulthood [25, 26], and regular physical activity has the potential to lower blood pressure and reduce BMI among adults [26–28].

There is a lack of information about the association between physical activity and blood pressure in children. Several studies have reported inverse associations between physical activity and systolic and/or diastolic blood pressure, with most showing this association is independent of BMI [29–40]. It is proposed that physical activity produces more favourable vascular health profiles among children [38, 41]. In a review of the literature published in 2007 [20], it was suggested that engaging in 40 minutes of moderate-to-vigorous-intensity physical activity (MVPA) on 3–5 days per week was required to improve vascular function and reduce blood pressure in obese children. However, other studies have reported no association between physical activity and blood pressure in children [24, 42–45]. The majority of studies to date were cross-sectional and used child or parental report of physical activity [31, 33, 36, 39, 41, 43, 45], which may be subject to misclassification, particularly in relation to physical activity intensity and duration. We previously published cross-sectional analyses using baseline data from the present study, finding no association between physical activity and sedentary time with blood pressure in 9-year-old children [24]. Therefore, there is a need for more prospective studies with measures of objectively-assessed physical activity and sedentary time to further understand key determinants of variation in child blood pressure.

In this study, we investigated the cross-sectional and prospective associations of BMI and accelerometer-assessed physical activity and sedentary time measured at age 9 and 11 with blood pressure at age 11.

## Methods

Data used in the present study were from B-PROACT1V, a longitudinal study examining physical activity and sedentary behaviours of children and parents as children progress

through primary school. The study has been described in detail elsewhere [46–49]. Briefly, in 2012–2013, data were collected from 1299 Year 1 children (median age: 6 years) who were recruited from 57 schools in Bristol, UK, and the surrounding area. In 2015–2016, data were collected from 1223 Year 4 children (median age: 9 years) from 47 of the original schools. In 2017–2018, 50 of the original schools were re-recruited and data were collected from 1296 Year 6 children (median age: 11 years). In total, 2132 children participated, of whom 958 participated at one time-point, 662 at two time-points, and 512 at three time-points. The current study used data from the 1283 children who provided at least two blood pressure measurements at age 11, since blood pressure was only measured at age 9 and 11, and the age 9 blood pressure data have been presented previously [24]. This included 797 children who also participated in the study at age 9 years. Ethical approval for the study was granted from the School for Policy Studies Research Ethics Committee at the University of Bristol, and written parent consent was provided for both child and parent participation.

## Blood pressure

Blood pressure was measured in the child's school by trained fieldworkers using an Omron 907 Professional Blood Pressure Monitor [50] with a small or medium cuff (OMRON Corporation, Kyoto, Japan). After three minutes of rest, measurements were taken, using the appropriate cuff size, three times, one-minute apart, in the left arm while the child was seated. Of the 1296 children who participated at age 11, we have included 1283 (99%) children who provided at least two blood pressure measurements at age 11, in the current study. The mean of all available blood pressure measurements was used in the analyses. In our main analyses we examined associations with difference in mean blood pressure as a continuous variable. We also examined associations with high systolic and diastolic blood pressure, defined as the systolic or diastolic blood pressure that was ≥95th percentile using US children's age, sex and height standardized blood pressure references [51]. US reference charts were used because of the lack of such population references for UK children.

## Body mass index

Children's weight and height were measured at the children's schools by trained fieldworkers. Weight was measured using a SECA 899 digital scale to the nearest 0.1kg, and height was measured using a SECA Leicester stadiometer to the nearest 0.1cm (HAB International, Northampton). BMI was derived as weight (kg)/height ($m^2$) and converted to an age- and sex-specific standard deviation score based on UK 1990 growth centiles [52].

## Accelerometer-assessed physical activity

MVPA and sedentary time of children were measured using ActiGraph wGT3X-BT (Pensacola, FL, USA) accelerometers. Children were asked to wear an accelerometer on their waist during all hours they were awake for five consecutive days, including weekend days. Accelerometer data were processed using Kinesoft (v3.3.75; Kinesoft, Saskatchewan, Canada). At least three valid days of data were required for inclusion in analysis, and a valid day was defined as ≥500 minutes of data, after excluding intervals of ≥60 minutes of zero counts (for the latter allowing up to two minutes of interruptions during the 60 minutes). The Evenson [53] population-specific cut points for children were used to derive the average number of MVPA and sedentary minutes per day.

## Observed confounders

Of the data available in this study we considered child's sex, age and height, household socio-economic position and parental history of high blood pressure as key confounders given their known influences on BMI, physical activity, sedentary behaviour and blood pressure. Dietary factors are likely to influence blood pressure, but we did not have information on this. Parental behaviours, such as smoking and alcohol consumption, may be mimicked by their children and hence they (or these behaviours in the child) may also confound the associations we are exploring. We considered that at age 9–11 years very few children would be smoking or drinking alcohol to an extent that would influence their blood pressure, and so did not consider these further. At least one of the children's parents were recruited to the study. Parents completed a questionnaire requesting information about their child's sex and date of birth. Where the child's date of birth was missing (8.4% of children), the median age was assigned (11.0 years at Year 6). While replacing missing data with average sample values can introduce bias, as children were from a single school year we felt this was appropriate. Parents were also asked if either of the child's biological parents had ever been informed that they had high blood pressure. As indicators of socioeconomic position, parents were asked to report the highest level of education in the home, with the following response options: 'up to GCSEs/ GCEs/ O Levels or equivalent' (qualifications usually obtained in several subjects at age 16, the minimum legal education leaving age in the UK up to 2015), 'A Levels/ NVQs/ GNVQs' (qualifications usually obtained at age 18), 'Degree/ Diploma/ HNC/ HND or equivalent' and 'Postgraduate degree (MSc, PhD)'. This was combined across time-points to produce a single indicator of highest household education.

## Statistical analysis

Means and proportions were used to examine the characteristics of the cross-sectional and prospective samples. Linear regression models were used to examine the cross-sectional associations between the child's BMI z-score, MVPA and sedentary time and systolic and diastolic blood pressure at age 11. Linear regression models were also used to examine the prospective associations between BMI, MVPA and sedentary time at age 9 and systolic and diastolic blood pressure at age 11. Model 1 was unadjusted and in the prospective models, we adjusted for baseline (age 9 years) levels of blood pressure (systolic or diastolic respectively). In Model 2, we additionally adjusted for highest household education, child age and height, child sex (for models with all children), parent reported high blood pressure, BMI z-score (for activity measures), and accelerometer wear time (for activity measures). Due to a previous study that used B-PROACT1V data finding a negative association between MVPA and BMI as children age [50], the models with activity measures were adjusted for BMI z-score to examine the independent effect of physical activity on blood pressure. Logistic regression models were used to explore cross-sectional and prospective associations with odds of having high systolic and diastolic blood pressure. We undertook all analyses with girls and boys combined and explored differences between them by running analyses separately and exploring evidence for statistical interaction between sex and each exposure. To account for children being recruited via schools, robust standard errors, which took account of the school-level clustering, were used for all models. All analyses were performed using Stata version 15.0 (StataCorp, 2015).

**Dealing with missing data.** Among the 1283 children who were eligible for inclusion in the cross-sectional and the 797 eligible for the prospective analyses, there were small amounts of missing data for risk factors, and/or confounders (Table 1). This varied from 0 (e.g., for child blood pressure at age 11, child age and sex) to 17.2% (for parental high blood pressure reported at age 11) in the cross-sectional analyses and 14.8% (for parental high blood pressure

**Table 1. Characteristics of the children who participated in the study at age 11 years, and those who participated at both age 9 and age 11 years in the observed and multiple imputation data.**

| Characteristic | Children who participated at age 11 | | | Children who participated at ages 9 and 11 | | |
| --- | --- | --- | --- | --- | --- | --- |
| | Observed data | | Imputed (N = 1283) | Observed data | | Imputed (N = 797) |
| | N | Mean (SD) or % | Mean (SD) or % | N | Mean (SD) or % | Mean (SD) or % |
| Systolic blood pressure at age 11 (mm Hg) | 1283 | 104.65 (10.91) | 104.65 (10.91) | 797 | 104.60 (10.96) | 104.60 (10.96) |
| Diastolic blood pressure at age 11 (mm Hg) | 1283 | 68.70 (9.52) | 68.70 (9.52) | 797 | 68.72 (9.59) | 68.72 (9.59) |
| High systolic blood pressure at age 11 | 1283 | | | 797 | | |
| *No* | | 92.2% | 92.2% | | 92.3% | 92.3% |
| *Yes* | | 7.8% | 7.8% | | 7.7% | 7.7% |
| High diastolic blood pressure at age 11 | 1283 | | | 797 | | |
| *No* | | 89.3% | 89.3% | | 89.1% | 89.1% |
| *Yes* | | 10.7% | 10.7% | | 10.9% | 10.9% |
| Body mass index (z-score) at age 11 | 1279 | 0.35 (1.16) | 0.35 (1.16) | 795 | 0.32 (1.18) | 0.32 (1.18) |
| MVPA (mins/day) at age 11 | 1229 | 58.13 (22.52) | 58.11 (22.56) | 772 | 57.94 (22.38) | 57.84 (22.47) |
| Sedentary (mins/day) at age 11 | 1229 | 464.40 (68.66) | 464.49 (68.86) | 772 | 462.73 (66.07) | 462.79 (66.28) |
| Highest household education | 1180 | | | 773 | | |
| *Up to GCSE/O level* | | 20.5% | 20.8% | | 19.0% | 19.1% |
| *A level/ NVQ* | | 26.2% | 26.3% | | 26.0% | 26.0% |
| *Degree/ HND* | | 36.6% | 36.3% | | 37.8% | 37.7% |
| *Higher degree (MSc/PhD)* | | 16.7% | 16.6% | | 17.2% | 17.2% |
| Parent/s had high blood pressure | 1062 | | | 679 | | |
| *No* | | 83.9% | 83.3% | | 84.5% | 83.7% |
| *Yes* | | 16.1% | 16.7% | | 15.5% | 16.3% |
| Systolic blood pressure at age 9 (mm Hg) | - | - | - | 786 | 106.22 (12.12) | 106.05 (11.70) |
| Diastolic blood pressure at age 9 (mm Hg) | - | - | - | 786 | 70.64 (11.17) | 70.50 (10.74) |
| High systolic blood pressure at age 9 | | | | 786 | | |
| *No* | - | - | - | | 86.1% | 86.5% |
| *Yes* | - | - | - | | 13.9% | 13.5% |
| High diastolic blood pressure at age 9 | | | | 786 | | |
| *No* | - | - | - | | 80.5% | 81.0% |
| *Yes* | - | - | - | | 19.5% | 19.0% |
| Body mass index (z-score) at age 9 | - | - | - | 796 | 0.29 (1.06) | 0.30 (1.06) |
| MVPA (mins/day) at age 9 | - | - | - | 760 | 62.44 (22.52) | 62.21 (22.57) |
| Sedentary (mins/day) at age 9 | - | - | - | 760 | 431.89 (60.44) | 433.22 (60.64) |

reported at age 9) in the prospective analyses. To enable us to include information from all study participants in our analysis, and thus potentially increase statistical power and minimise selection bias, we used multiple imputation of missing data using chained equations. Imputation was completed separately for cross-sectional and prospective analyses. Thus, for the cross-sectional analyses we imputed data for the 1283 children who participated at age 11 and provided at least two blood pressure measurements. For prospective analyses, which examined associations between BMI, physical activity or sedentary time at age 9 with blood pressure at age 11, we imputed to the 797 children who took part at both time points.

All child accelerometer measures, measurements of blood pressure, and characteristics that were potential predictors of missingness (child age, sex, BMI, highest household education, parental high blood pressure, and the child's school) at either year, were included in the multiple imputation models. Children's classification of high systolic and diastolic blood pressure were

imputed passively. Twenty imputed datasets were created using 20 cycles of regression switching and combined regression coefficients across imputed datasets using Rubin's rules [54].

Regression analyses were repeated restricting to children who had complete data on all exposures, outcomes and covariables in cross-sectional and prospective analyses (N = 1025 and 655 respectively). The results were very similar between the main analyses with multiple imputed datasets and the complete case analyses for both cross-sectional and prospective analyses and, therefore, the results of the complete case analyses are not presented but are available from the authors on request.

## Results

The characteristics of children who provided at least two measures of blood pressure at age 11, and the subset who additionally took part at age 9, in the observed and imputed datasets are shown in Table 1. The subset that took part in both years were comparable to the whole age 11 sample, and in both years the distributions of all characteristics were very similar in imputed and observed data. The mean systolic blood pressure for 11-year-old children was 104.65 mm Hg (7.8% had high systolic blood pressure) and mean diastolic blood pressure was 68.70 mm Hg (10.7% had high diastolic blood pressure).

### BMI and blood pressure

The cross-sectional associations of BMI with systolic and diastolic blood pressure at age 11 are shown in Table 2. A one standard deviation increase in BMI was associated with increases of 0.91 and 1.08 mm Hg for systolic and diastolic blood pressure, respectively, in the confounder-adjusted models. The positive associations between BMI and systolic blood pressure were stronger among boys than girls. There was also some evidence to suggest that BMI at age 11

**Table 2. Cross-sectional associations of body mass index, physical activity and sedentary time with blood pressure at age 11 in the imputed data (N = 1283).**

| Exposure | Systolic blood pressure (mm Hg) at age 11 | | | Diastolic blood pressure (mm Hg) at age 11 | | |
|---|---|---|---|---|---|---|
| | Coefficient (95% Confidence Interval) | | | Coefficient (95% Confidence Interval) | | |
| | All (N = 1283) | Boys (N = 611) | Girls (N = 672) | All (N = 1283) | Boys (N = 611) | Girls (N = 672) |
| **BMI z-score at age 11 (per SD of BMI)** | | | | | | |
| Model 1 | 1.30 (0.75 to 1.84) | 1.53 (0.66 to 2.39) | 1.09 (0.36 to 1.82) | 1.15 (0.67 to 1.63) | 1.11 (0.28 to 1.94) | 1.20 (0.66 to 1.74) |
| Model 2 | 0.91 (0.32 to 1.50) | 1.21 (0.28 to 2.15) | 0.62 (-0.19 to 1.42) | 1.08 (0.54 to 1.62) | 1.10 (0.21 to 2.00) | 1.00 (0.35 to 1.64) |
| P value for sex interaction | 0.43 | | | 0.99 | | |
| **MVPA at age 11 (per 10 mins/day)** | | | | | | |
| Model 1 | 0.07 (-0.25 to 0.38) | 0.12 (-0.28 to 0.52) | -0.06 (-0.48 to 0.36) | -0.15 (-0.45 to 0.15) | 0.06 (-0.33 to 0.45) | -0.26 (-0.66 to 0.14) |
| Model 2 | 0.15 (-0.17 to 0.47) | 0.29 (-0.10 to 0.68) | -0.03 (-0.45 to 0.40) | 0.05 (-0.24 to 0.33) | 0.19 (-0.18 to 0.56) | -0.14 (-0.55 to 0.26) |
| P value for sex interaction | 0.45 | | | 0.23 | | |
| **Sedentary time at age 11 (per 10 mins/day)** | | | | | | |
| Model 1 | -0.03 (-0.15 to 0.09) | -0.08 (-0.22 to 0.06) | 0.03 (-0.13 to 0.19) | 0.02 (-0.07 to 0.12) | -0.01 (-0.12 to 0.10) | 0.04 (-0.10 to 0.18) |
| Model 2 | -0.01 (-0.18 to 0.15) | -0.08 (-0.29 to 0.13) | 0.05 (-0.14 to 0.23) | 0.08 (-0.05 to 0.21) | 0.01 (-0.16 to 0.17) | 0.15 (-0.03 to 0.33) |
| P value for sex interaction | 0.34 | | | 0.56 | | |

Model 1 is adjusted for clustering at the school level; Model 2 is additionally adjusted for child age and height at the age 11 data collection, child sex (for models with all children), highest household education, parental high blood pressure, BMI z-score (for activity variables) and accelerometer wear time (for activity variables).

was associated with increased odds of having high systolic blood pressure among boys, but not in girls or in the overall sample (Table 3). A one standard deviation increase in BMI was associated with 27% increased odds of high diastolic blood pressure in the overall sample. Evidence for associations between BMI and odds of having high diastolic blood pressure were stronger among girls than boys.

In the prospective models, BMI at age 9 was positively associated with diastolic blood pressure at age 11 in the overall sample and for girls, but not boys (Table 4). A one standard deviation increase in BMI at age 9 was associated with a 1.36 mm Hg increase in diastolic blood pressure at age 11 for girls. There was some evidence for a positive association between BMI at age 9 and systolic blood pressure at age 11 in the unadjusted model, but this association was not evident in the adjusted model. In the prospective models for odds of having high blood pressure, BMI at age 9 was associated with increased odds of having high diastolic blood pressure at age 11 in the overall sample and for girls (30% and 32% respectively, Table 5). There was no evidence for associations between BMI at age 9 and odds of having high systolic blood pressure at age 11, or with high diastolic blood pressure at age 11 among boys.

## Physical activity and blood pressure

In the cross-sectional models, there was no strong evidence for associations between MVPA or sedentary time with systolic or diastolic blood pressure (Table 2), or odds of having high systolic or diastolic blood pressure (Table 3), in any of the models. In the prospective analyses, there was weak evidence for a positive association between sedentary time at age 9 and diastolic blood pressure at age 11, but this association was only evident in the unadjusted model (Table 4). For girls, MVPA at age 9 was associated with increased odds of having high diastolic blood pressure at age 11 in the unadjusted model only (Table 5). Similarly, there was evidence for a positive association between sedentary time at age 9 and odds of having high systolic blood pressure at age 11 among girls in both models. There was also weak evidence (unadjusted model only) that sedentary time at age 9 was associated with increased odds of having high diastolic blood pressure at age 11 in the overall sample and among girls, but not boys.

**Table 3. Cross-sectional associations of body mass index, physical activity and sedentary time with odds of high systolic and diastolic blood pressure at age 11 in the imputed data (N = 1283).**

| Exposure | High systolic blood pressure at age 11 | | | High diastolic blood pressure at age 11 | | |
|---|---|---|---|---|---|---|
| | Odds Ratio (95% Confidence Interval) | | | Odds Ratio (95% Confidence Interval) | | |
| | All (N = 1283) | Boys (N = 611) | Girls (N = 672) | All (N = 1283) | Boys (N = 611) | Girls (N = 672) |
| **BMI z-score at age 11 (per SD of BMI)** | | | | | | |
| Model 1 | 1.13 (0.98 to 1.30) | 1.20 (0.96 to 1.51) | 1.08 (0.89 to 1.30) | 1.25 (1.06 to 1.47) | 1.21 (0.92 to 1.60) | 1.28 (1.06 to 1.54) |
| Model 2 | 1.13 (0.97 to 1.32) | 1.32 (1.06 to 1.66) | 1.00 (0.80 to 1.24) | 1.27 (1.05 to 1.53) | 1.28 (0.94 to 1.73) | 1.24 (1.01 to 1.53) |
| P value for sex interaction | 0.38 | | | 0.84 | | |
| **MVPA at age 11 (per 10 mins/day)** | | | | | | |
| Model 1 | 0.94 (0.85 to 1.04) | 0.99 (0.88 to 1.12) | 0.88 (0.76 to 1.03) | 0.96 (0.89 to 1.03) | 1.01 (0.90 to 1.12) | 0.96 (0.84 to 1.08) |
| Model 2 | 0.96 (0.86 to 1.06) | 1.02 (0.90 to 1.17) | 0.90 (0.77 to 1.05) | 1.01 (0.94 to 1.10) | 1.05 (0.93 to 1.17) | 0.98 (0.87 to 1.12) |
| P value for sex interaction | 0.26 | | | 0.58 | | |
| **Sedentary time at age 11 (per 10 mins/day)** | | | | | | |
| Model 1 | 1.01 (0.97 to 1.04) | 1.00 (0.96 to 1.05) | 1.01 (0.96 to 1.05) | 0.99 (0.97 to 1.02) | 0.99 (0.96 to 1.02) | 0.99 (0.96 to 1.03) |
| Model 2 | 1.03 (0.98 to 1.08) | 1.01 (0.95 to 1.08) | 1.04 (0.98 to 1.10) | 1.01 (0.97 to 1.04) | 1.00 (0.95 to 1.05) | 1.01 (0.96 to 1.06) |
| P value for sex interaction | 0.98 | | | 0.97 | | |

Model 1 is adjusted for clustering at the school level; Model 2 is additionally adjusted for child age and height at the age 11 data collection, child sex (for models with all children), highest household education, parental high blood pressure, BMI z-score (for activity variables) and accelerometer wear time (for activity variables).

**Table 4. Prospective associations of body mass index, physical activity and sedentary time at age 9 with blood pressure at age 11 in the imputed data (N = 797).**

| Exposure | Systolic blood pressure (mm Hg) at age 11 | | | Diastolic blood pressure (mm Hg) at age 11 | | |
|---|---|---|---|---|---|---|
| | Coefficient (95% Confidence Interval) | | | Coefficient (95% Confidence Interval) | | |
| | All (N = 797) | Boys (N = 355) | Girls (N = 442) | All (N = 797) | Boys (N = 355) | Girls (N = 442) |
| **BMI z-score at age 9 (per SD of BMI)** | | | | | | |
| Model 1 | 0.98 (0.16 to 1.80) | 1.05 (-0.19 to 2.29) | 0.90 (-0.03 to 1.84) | 1.18 (0.62 to 1.74) | 0.83 (-0.12 to 1.78) | 1.44 (0.77 to 2.10) |
| Model 2 | 0.71 (-0.13 to 1.56) | 0.79 (-0.55 to 2.12) | 0.57 (-0.42 to 1.55) | 1.16 (0.49 to 1.84) | 0.85 (-0.24 to 1.93) | 1.36 (0.59 to 2.13) |
| P value for sex interaction | 0.70 | | | 0.33 | | |
| **MVPA at age 9 (per 10 mins/day)** | | | | | | |
| Model 1 | 0.24 (-0.11 to 0.59) | 0.30 (-0.22 to 0.82) | 0.24 (-0.25 to 0.74) | -0.02 (-0.33 to 0.29) | -0.004 (-0.46 to 0.45) | 0.20 (-0.37 to 0.76) |
| Model 2 | 0.22 (-0.15 to 0.59) | 0.32 (-0.19 to 0.84) | 0.13 (-0.35 to 0.61) | 0.03 (-0.28 to 0.35) | 0.008 (-0.44 to 0.46) | 0.09 (-0.41 to 0.59) |
| P value for sex interaction | 0.73 | | | 0.85 | | |
| **Sedentary time at age 9 (per 10 mins/day)** | | | | | | |
| Model 1 | 0.03 (-0.09 to 0.15) | -0.06 (-0.24 to 0.12) | 0.12 (-0.05 to 0.29) | 0.12 (0.002 to 0.24) | 0.09 (-0.08 to 0.26) | 0.13 (-0.03 to 0.29) |
| Model 2 | -0.02 (-0.21 to 0.17) | -0.11 (-0.38 to 0.15) | 0.08 (-0.16 to 0.33) | 0.06 (-0.14 to 0.25) | 0.04 (-0.23 to 0.31) | 0.09 (-0.14 to 0.32) |
| P value for sex interaction | 0.23 | | | 0.54 | | |

Model 1 is adjusted for systolic and diastolic blood pressure (respectively) at age 9 and for clustering at the school level; Model 2 is additionally adjusted for child age and height at the age 9 data collection, child sex (for models with all children), highest household education, parental high blood pressure, BMI z-score at age 9 (for activity variables) and accelerometer wear time at age 9 (for activity variables).

**Table 5. Prospective associations of body mass index, physical activity and sedentary time at age 9 with odds of high systolic and diastolic blood pressure at age 11 in the imputed data (N = 797).**

| Exposure | High systolic blood pressure at age 11 | | | High diastolic blood pressure at age 11 | | |
|---|---|---|---|---|---|---|
| | Odds Ratio (95% CI) | | | Odds Ratio (95% CI) | | |
| | All (N = 797) | Boys (N = 355) | Girls (N = 442) | All (N = 797) | Boys (N = 355) | Girls (N = 442) |
| **BMI z-score at age 9 (per SD of BMI)** | | | | | | |
| Model 1 | 1.12 (0.89 to 1.40) | 1.06 (0.81 to 1.39) | 1.15 (0.83 to 1.58) | 1.30 (1.10 to 1.53) | 1.15 (0.84 to 1.58) | 1.38 (1.13 to 1.67) |
| Model 2 | 1.11 (0.86 to 1.43) | 1.12 (0.83 to 1.53) | 1.07 (0.76 to 1.51) | 1.30 (1.06 to 1.59) | 1.20 (0.82 to 1.76) | 1.32 (1.04 to 1.68) |
| P value for sex interaction | 0.79 | | | 0.53 | | |
| **MVPA at age 9 (per 10 mins/day)** | | | | | | |
| Model 1 | 1.04 (0.92 to 1.17) | 1.15 (0.92 to 1.43) | 0.97 (0.81 to 1.15) | 1.04 (0.92 to 1.16) | 0.98 (0.80 to 1.20) | 1.17 (1.00 to 1.38) |
| Model 2 | 1.04 (0.91 to 1.19) | 1.17 (0.95 to 1.44) | 0.92 (0.78 to 1.08) | 1.06 (0.95 to 1.19) | 0.99 (0.83 to 1.19) | 1.13 (0.97 to 1.31) |
| P value for sex interaction | 0.19 | | | 0.27 | | |
| **Sedentary time at age 9 (per 10 mins/day)** | | | | | | |
| Model 1 | 1.03 (0.99 to 1.07) | 0.99 (0.93 to 1.05) | 1.07 (1.02 to 1.13) | 1.03 (1.00 to 1.06) | 1.01 (0.96 to 1.06) | 1.04 (1.00 to 1.07) |
| Model 2 | 1.02 (0.96 to 1.09) | 0.96 (0.86 to 1.06) | 1.08 (1.01 to 1.16) | 1.00 (0.94 to 1.06) | 1.01 (0.92 to 1.10) | 1.00 (0.93 to 1.07) |
| P value for sex interaction | 0.03 | | | 0.32 | | |

Model 1 is adjusted for systolic and diastolic blood pressure (respectively) at age 9 and for clustering at the school level; Model 2 is additionally adjusted for child age and height at the age 9 data collection, child sex (for models with all children), highest household education, parental high blood pressure, BMI z-score at age 9 (for activity variables) and accelerometer wear time at age 9 (for activity variables).

## Discussion

In this study, we found small, but consistent, cross-sectional and prospective associations of higher BMI with higher mean diastolic blood pressure and the likelihood of having high diastolic blood pressure. Prospectively, a one standard deviation increase in BMI at age 9 was associated with an increase of 1.16 mm Hg for diastolic blood pressure at age 11, as well as 30% increased odds of high diastolic blood pressure at that age. These associations were stronger in girls compared to boys. For systolic blood pressure, the association with BMI was only evident in the cross-sectional models at age 11. There was only very weak evidence that children's physical activity or sedentary time at age 9 were prospectively associated with blood pressure at age 11 years. These findings suggest that while greater BMI during middle childhood may influence the future risk of higher diastolic blood pressure, interventions aimed at increasing physical activity and reducing sedentary time are unlikely to impact the development of cardiovascular disease risk during childhood. Whereas, interventions to reduce BMI, or prevent high BMI, have the potential to impact children's blood pressure. The findings also suggest that childhood BMI might be a more important risk factor for higher diastolic than systolic blood pressure.

This study adds prospective evidence, with blood pressure measured at two time-points, and therefore the ability to adjust for baseline blood pressure, to the existing cross-sectional and prospective studies suggesting that there is a positive association between BMI and blood pressure among children [15–24]. A cross-sectional study with 3923 children aged 6–11 years from southern Italy found BMI and waist circumference z-scores were positively associated with blood pressure [16]. Another cross-sectional study of 1432 twelve-year-olds found a high BMI and large waist circumference (above the 90th percentile) were associated with higher systolic and diastolic blood pressure levels and adverse blood cholesterol levels [22]. These studies highlight the potential usefulness of both BMI and waist circumference measures for identifying those at risk of future adverse cardiovascular risk profiles. A large prospective study of 5235 children also from Bristol, UK (Avon Longitudinal Study of Parents and Children (ALSPAC)), found a one standard deviation (SD) increase in BMI at age 9–12 years was associated with an increased risk of high systolic blood pressure (≥130mm Hg) at age 15–16 in girls (odds ratio (OR): 1.23, 95% confidence interval (CI): 1.10 to 1.38) and boys (OR: 1.24, 95% CI: 1.13 to 1.37) [21]. However, they found no evidence for associations between BMI and high diastolic blood pressure [21], which is contrary to our findings at somewhat younger ages of an association with diastolic but not systolic blood pressure. This difference may be due to chance or to differences in age or birth cohort; children in the current cohort were born in 2006–2007, while the ALSPAC participants were born in the 1990s, when population levels of childhood obesity were lower [55]. The prospective associations between BMI with high diastolic blood pressure in the present study were greater in magnitude than the previously published study of cross-sectional associations at age 9 in the same cohort (B-PROACT1V), using the equivalent definition of high blood pressure and the same confounders [24]. This suggests that excessive weight during childhood may be progressively associated with risk of elevated diastolic blood pressure as children age, and this may be particularly the case for girls. With replication and additional causal evidence, for example from methods such as within sibship analyses or Mendelian randomization, this would suggest that effective obesity prevention interventions, especially those that target girls, are needed from an early age. However, it is also possible that the somewhat weaker association in the earlier cross-sectional analyses are chance findings.

In the current study, only one physical activity or sedentary time variable was associated with blood pressure in the adjusted models. For girls, sedentary time at age 9 was positively associated with the odds of having high systolic blood pressure at age 11 (OR: 1.08, 95% CI:

1.01 to 1.16), but the evidence of association was weak and there was no association for boys, in the overall sample, or with blood pressure measured as a continuous variable. In contrast to our lack of association between physical activity and blood pressure, a study from Birmingham, UK (cross-sectional analyses: N = 512; two-year follow-up prospective analyses: N = 427), found total physical activity was inversely associated with diastolic blood pressure cross-sectionally and prospectively; there was not strong evidence of association with systolic blood pressure in either analysis [38]. Similarly, a cross-sectional study examining the cardio-vascular health of 2049 9-to-10-year-old children from three UK cities, found total physical activity level was inversely associated with lower diastolic blood pressure, but no evidence for an association with systolic blood pressure [30]. The contrast in findings between these studies and the current study may be due to the differences between cohorts. The Birmingham participants were younger at baseline (mean age 6.5 years, range 5.4–7.8 years) and predominantly from a South Asian background [38], and the multi-city sample were also more ethnically diverse [30] compared to the current study. In an earlier B-PROACT1V study [24], there was no evidence that physical activity or sedentary time at 6 or 9 years were cross-sectionally or prospectively associated with systolic or diastolic blood pressure at 9 years. The present study adds to this evidence, demonstrating a lack of strong evidence of association between physical activity and sedentary time with blood pressure at 11 years within the same cohort. These findings are in line with a Danish study that examined the cross-sectional associations between objectively-assessed physical activity and metabolic syndrome among 589 8–10 year-old-children [44]. The study found no evidence of an association between physical activity and systolic or diastolic blood pressure [44]. The conflicting findings in the literature suggest that other risk factors for higher blood pressure (e.g., BMI, genetic influences) may be more important than physical activity in primary school aged children. They also suggest that interventions to reduce BMI via physical activity and sedentary behaviour may not directly reduce blood pressure in children.

In adults, a 2 mm Hg reduction in blood pressure is associated with a 6% reduction in coronary heart disease and a 15% reduction in stroke-related events [56]. In the present study, the cross-sectional differences in systolic and diastolic blood pressure per standard deviation of BMI were small (0.91 and 1.08 mm Hg, respectively), suggesting that relatively large reductions in BMI are needed to reduce cardiovascular disease risk. The findings do indicate that measuring BMI in primary-school children is a suitable and relatively low-cost measure (compared to estimating body composition using dual energy x-ray absorptiometry) for identifying those at risk of future adverse cardiovascular risk profiles. Prevention strategies are needed to shift the population distribution of childhood adiposity downwards.

## Strengths and limitations

Strengths of this study include the measurement of blood pressure at two ages in childhood, and the objective measurements of BMI and physical activity (via accelerometers), allowing us to examine cross-sectional and prospective associations of these exposures at ages 9 and 11 years with blood pressure at age 11. Multiple imputation of missing data was used to increase precision and reduce selection bias in our coefficient estimates [57]. Findings in the imputed data were consistent with the complete case analyses. Due to the observational nature of the study, we were unable to exclude residual confounding, for example by dietary factors such as salt, sugar or fat intake. Low birth weight is also associated with high blood pressure later in life, but this information was not available in the present study [58]. The study sample was relatively homogenous, primarily of White British origin from one area of the UK, which limits the ability to extrapolate to other ethnic groups in more diverse areas of the UK.

## Conclusions

The findings of the present study strengthen existing evidence suggesting that BMI may be a risk factor in the development of high diastolic blood pressure during childhood. Conversely, the amount of time that predominantly White British primary school aged children spend being physically active or sedentary does not appear to be strongly associated with blood pressure. These results suggest that interventions to prevent excessive bodyweight may be important in the prevention of cardiovascular disease risk during childhood. Current evidence is limited on the effectiveness of physical activity interventions on BMI [59], and our results suggest targeting this may not directly reduce blood pressure, therefore, future obesity prevention initiatives should target multiple components (e.g., physical activity, nutrition, and emotional well-being), rather than focus on increasing physical activity in isolation.

## Acknowledgments

We would like to thank all of the families and schools that have taken part in the B-PROACT1V project. We would also like to thank all current and previous members of the research team who are not authors on this paper.

## Author Contributions

**Conceptualization:** Janice L. Thompson, Simon J. Sebire, Deborah A. Lawlor, Russell Jago.

**Formal analysis:** Emma Solomon-Moore, Ruth Salway.

**Funding acquisition:** Janice L. Thompson, Simon J. Sebire, Deborah A. Lawlor, Russell Jago.

**Investigation:** Emma Solomon-Moore, Lydia Emm-Collison.

**Project administration:** Emma Solomon-Moore, Lydia Emm-Collison.

**Supervision:** Russell Jago.

**Writing – original draft:** Emma Solomon-Moore.

**Writing – review & editing:** Ruth Salway, Lydia Emm-Collison, Janice L. Thompson, Simon J. Sebire, Deborah A. Lawlor, Russell Jago.

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
