## [Decision Letter · Decision Letter 0]

13 Jan 2020

PONE-D-19-27408

Associations of body mass index, physical activity and sedentary time with blood pressure in primary school children from south-west England: a prospective study

PLOS ONE

Dear Dr Emm-Collison,

Thank you for submitting your manuscript to PLOS ONE. After careful consideration, we feel that it has merit but does not fully meet PLOS ONE’s publication criteria as it currently stands. Therefore, we invite you to submit a revised version of the manuscript that addresses the points raised during the review process.

Please improve analyses and revise manuscript accordingly.

We would appreciate receiving your revised manuscript by Feb 27 2020 11:59PM. To enhance the reproducibility of your results, we recommend that if applicable you deposit your laboratory protocols in protocols.io, where a protocol can be assigned its own identifier (DOI) such that it can be cited independently in the future. For instructions see: http://journals.plos.org/plosone/s/submission-guidelines#loc-laboratory-protocols

We look forward to receiving your revised manuscript.

Kind regards,

Guoying Wang, MD, PhD

Academic Editor

PLOS ONE

Journal Requirements:

2. We note that your study is closely related to the following publication, on which you are an author:

https://journals.plos.org/plosone/article?id=10.1371/journal.pone.0188618

Please ensure you cite and discuss the above study in the introduction and discussion section of your manuscript, clarifying how the present work is related to the previously published paper.

Please note that our second publication criterion states that "If a submitted study replicates or is very similar to previous work, authors must provide a sound scientific rationale for the submitted work and clearly reference and discuss the existing literature. Submissions that replicate or are derivative of existing work will likely be rejected if authors do not provide adequate justification." http://www.plosone.org/static/publication.action#results.

Thank you for your attention to this request.

Reviewers' comments:

Reviewer's Responses to Questions

**Comments to the Author**

1. Is the manuscript technically sound, and do the data support the conclusions?

Reviewer #1: Yes

Reviewer #2: Partly

2. Has the statistical analysis been performed appropriately and rigorously? 

Reviewer #1: Yes

Reviewer #2: No

3. Have the authors made all data underlying the findings in their manuscript fully available?

Reviewer #1: Yes

Reviewer #2: No

4. Is the manuscript presented in an intelligible fashion and written in standard English?

Reviewer #1: Yes

Reviewer #2: Yes

5. Review Comments to the Author

Reviewer #1: The authors studied the associations between body composition, physical activity and sedentary time with blood

pressure in English children at the primary school stage. The manuscript is well written. The language is clear and straight forward. The methodology is fine. The argument is robust and the conclusion is based on the findings.

Reviewer #2: This study aims to investigate the associations of BMI and physical activity and sedentary time with blood pressure at age 11 cross-sectionally and prospectively. Authors found that BMI was associated with diastolic blood pressure in both cross-sectional and prospective analyses, but not for physical activity and sedentary time. They concluded that “These results suggest that interventions to prevent excessive bodyweight may be important in the prevention of cardiovascular disease risk during childhood. Current evidence is limited on the effectiveness of physical activity interventions on BMI, and our results suggest targeting this may not directly reduce blood pressure, therefore, future obesity prevention initiatives should target multiple components (e.g., physical activity, nutrition, and emotional well-being), rather than focus on increasing physical activity in isolation.” This study is well written, but additional analysis is needed.

Comments:

1. It is well recognized that less physical activity and overnutrition contribute to childhood overweight or obesity. Therefore, it is not reasonable to analyze physical activity or sedentary time without taking into account BMI (or overweight or obesity). Additional analysis is needed to rerun the regression models between physical activity and sedentary time and blood pressure (BP) stratified by BMI categories, and examine the combined effect of physical activity or sedentary time with BMI on BP.

2. Reviewer is interested if there is any relationship between physical activity and sedentary time with BMI in this study population.

3. Based on Barker hypothesis, low birthweight is associated with high BP in later life. It is unclear if birthweight is available in this study.

4. In Table 3, in the top row, left was systolic BP, right should be diastolic BP. Please check.

5. In Table 5, the odds ratio is 1.30 for overall sample, but 1.32 for girls. Therefore, in Page 18, lines 300-302, the sentence “Prospectively, a one standard deviation increase in BMI at age 9 was associated with an increase of 1.16 mm Hg for diastolic blood pressure at age 11, as well as 32% increased odds of high diastolic blood pressure at that age.” should be “ …….., as well as 30% increased .........”

6. Since no data show the relationship between physical activity and sedentary time with BMI in this study population, the results of this study do not support the statement in Page 19, lines 306-310: “These findings suggest that while greater BMI during the latter stages of primary school may influence the future risk of higher diastolic blood pressure, interventions aimed at reducing BMI via increased physical activity and reduced sedentary time are unlikely to impact the development of cardiovascular disease risk during childhood.”

7. Introduction section could be more concise.

6. PLOS authors have the option to publish the peer review history of their article (what does this mean?). If published, this will include your full peer review and any attached files.

Reviewer #1: No

Reviewer #2: No

---

## [Author Response · Author response to Decision Letter 0]

13 Feb 2020

Dear PLOS ONE Editorial Board,

Thank you for the opportunity to revise and resubmit the manuscript ‘Associations of body mass index, physical activity and sedentary time with blood pressure in primary school children from south-west England: a prospective study’. We have now amended the document based on the reviewer reports, we thank them for their supportive and constructive comments on the paper. Please find below a list of the editor and reviewer comments and our responses to each item. We have highlighted the changes in the manuscript in yellow.

Yours faithfully,

Lydia Emm-Collison

Comments from the editor:

Response: The manuscript formatting has been edited to meet PLOS ONE’s style requirements.

2. We note that your study is closely related to the following publication, on which you are an author: https://journals.plos.org/plosone/article?id=10.1371/journal.pone.0188618. Please ensure you cite and discuss the above study in the introduction and discussion section of your manuscript, clarifying how the present work is related to the previously published paper.

Response: Thank you for this advice, we have now made it clearer in the introduction (lines 87-92) and discussion (lines 321-324) that the previous publication was based on an earlier dataset (blood pressure measured at age 9) from the same longitudinal cohort study, and that this paper adds prospective evidence with blood pressure measured at two time points (age 9 and age 11) at a later stage in childhood compared to the previous study.

Response: The dataset has now been uploaded to an approved repository (Zenodo). The DOI is as follows: 10.5281/zenodo.1049587

Reviewer #1: 

The authors studied the associations between body composition, physical activity and sedentary time with blood pressure in English children at the primary school stage. The manuscript is well written. The language is clear and straight forward. The methodology is fine. The argument is robust and the conclusion is based on the findings.

Response: We thank the reviewer for the positive and supportive comments on the manuscript.

Reviewer #2: 

This study aims to investigate the associations of BMI and physical activity and sedentary time with blood pressure at age 11 cross-sectionally and prospectively. Authors found that BMI was associated with diastolic blood pressure in both cross-sectional and prospective analyses, but not for physical activity and sedentary time. They concluded that “These results suggest that interventions to prevent excessive bodyweight may be important in the prevention of cardiovascular disease risk during childhood. Current evidence is limited on the effectiveness of physical activity interventions on BMI, and our results suggest targeting this may not directly reduce blood pressure, therefore, future obesity prevention initiatives should target multiple components (e.g., physical activity, nutrition, and emotional well-being), rather than focus on increasing physical activity in isolation.” This study is well written, but additional analysis is needed.

Response: We thank the reviewer for their supportive comments on the manuscript. We have responded to each of their comments below.

Comments:

1. It is well recognized that less physical activity and overnutrition contribute to childhood overweight or obesity. Therefore, it is not reasonable to analyze physical activity or sedentary time without taking into account BMI (or overweight or obesity). Additional analysis is needed to rerun the regression models between physical activity and sedentary time and blood pressure (BP) stratified by BMI categories, and examine the combined effect of physical activity or sedentary time with BMI on BP.

Response: We agree that physical activity is associated with children’s body mass index and that is why the models for MVPA and sedentary time are adjusted for children’s BMI z-score. This takes into account BMI as a potential confounder when we analysed the association between physical activity and sedentary time with blood pressure. We have made this clearer in the methods (lines 175-178). Stratification by BMI categories would only be appropriate if BMI was a moderator (i.e., different relationships between blood pressure and physical activity depending on BMI), but there is no evidence to believe this is the case and investigating this is not a focus of the current manuscript. Therefore, we have not stratified the analyses by BMI categories.

2. Reviewer is interested if there is any relationship between physical activity and sedentary time with BMI in this study population.

Response: We have now added a brief section to the methods (lines 175-178) to highlight the negative association between MVPA and BMI as children age within this study population. This data has been published separately and we provide the citation for readers to obtain more information if they wish.

3. Based on Barker hypothesis, low birthweight is associated with high BP in later life. It is unclear if birthweight is available in this study.

Response: Unfortunately, information on birthweight was not available in the current study. We have added a sentence to the limitations section (lines 378-380) to explain this association and state that this data was unavailable.

4. In Table 3, in the top row, left was systolic BP, right should be diastolic BP. Please check.

Response: Thank you for spotting this error, this has now been amended (line 241).

5. In Table 5, the odds ratio is 1.30 for overall sample, but 1.32 for girls. Therefore, in Page 18, lines 300-302, the sentence “Prospectively, a one standard deviation increase in BMI at age 9 was associated with an increase of 1.16 mm Hg for diastolic blood pressure at age 11, as well as 32% increased odds of high diastolic blood pressure at that age.” should be “ …….., as well as 30% increased .........”

Response: This has now been amended (line 290).

6. Since no data show the relationship between physical activity and sedentary time with BMI in this study population, the results of this study do not support the statement in Page 19, lines 306-310: “These findings suggest that while greater BMI during the latter stages of primary school may influence the future risk of higher diastolic blood pressure, interventions aimed at reducing BMI via increased physical activity and reduced sedentary time are unlikely to impact the development of cardiovascular disease risk during childhood.”

Response: In response to the reviewer’s earlier comment, we now mention the negative association between physical activity and BMI in this study population within the methods section (lines 175-178). Regardless, we have edited the sentence to ensure that the results of the present study support the statement (lines 294-299):

“These findings suggest that while greater BMI during middle childhood may influence the future risk of higher diastolic blood pressure, interventions aimed at increasing physical activity and reducing sedentary time are unlikely to impact the development of cardiovascular disease risk during childhood. Whereas, interventions to reduce BMI, or prevent high BMI, have the potential to impact children’s blood pressure.”

7. Introduction section could be more concise.

Response: We agreed and responded by cutting several sentences out of the introduction that were deemed repetitive or unnecessary, and so we believe the introduction is now more concise. We have highlighted in the text where these sentences were cut from (lines 55, 57, 80).

---

## [Editor Report · Decision Letter 1]

14 Apr 2020

Associations of body mass index, physical activity and sedentary time with blood pressure in primary school children from south-west England: a prospective study

PONE-D-19-27408R1

Dear Dr. Dr Lydia Emm-Collison,

We are pleased to inform you that your manuscript has been judged scientifically suitable for publication and will be formally accepted for publication once it complies with all outstanding technical requirements.

With kind regards,

Pedro Tauler, Ph.D.

Academic Editor

PLOS ONE
---

## [Editor Report · Acceptance letter]

17 Apr 2020

PONE-D-19-27408R1 

Associations of body mass index, physical activity and sedentary time with blood pressure in primary school children from south-west England: a prospective study 

Dear Dr. Emm-Collison:

I am pleased to inform you that your manuscript has been deemed suitable for publication in PLOS ONE. Congratulations! Your manuscript is now with our production department. 

With kind regards,

on behalf of

Dr. Pedro Tauler 

Academic Editor

PLOS ONE